

# Multiple same-level and telescoping nesting in GFDL's dynamical core

Joseph Mouallem[1,2], Lucas Harris[2], and Rusty Benson[2]

[1]Cooperative Institute for Modeling the Earth System, Program in Oceanic and Atmospheric Sciences, Princeton University, Princeton, NJ, USA
[2]NOAA/Geophysical Fluid Dynamics Laboratory, Princeton, NJ, USA

**Correspondence:** J. Mouallem (mouallem@princeton.edu)

**Abstract.** Two way multiple same-level and telescoping grid nesting capabilities are implemented in the Geophysical Fluid Dynamics Laboratory (GFDL)'s Finite-Volume Cubed Sphere Dynamical core (FV3). Simulations are performed within GFDL's System for High-resolution modeling for Earth-to-Local Domains (SHiELD) using global and regional multiple nests configuration. Results show that multiple same level and multi-level telescoping nests were able capture various weather events in greater details by resolving smaller scale flow structures. Two-way updates do not introduce numerical errors in their corresponding parent grids where the nests are located. The cases of Hurricane Laura's landfall and an atmospheric river in California were found to be more intense with increased levels of telescoping nesting. All nested grids run concurrently and adding additional nests with computer cores to a setup do not degrade the computational performance nor increase the simulation run time if the cores are optimally distributed among the grids.

## 1  Introduction

Resolving fine scale flow structures is necessary for an accurate forecast of special weather events such as severe storms and hurricanes. The multi-scale non-linear interaction of small scale features to large mesoscale structures affect the overall storm behavior. Thus, investments in developing high resolution models by many organizations, scientists and research laboratories have become crucial to advance our understanding of such phenomena. Indeed, remarkable progress has been made over the last decade by the research community.

Running high resolution global models on grids fine enough to accurately capture special weather events such as hurricanes is still computationally expensive even with today's latest supercomputers. Regional or limited-area models, on the other hand, present limitations due to the handling of boundary conditions that could lead to propagation of numerical errors in the simulations and thus affect the accuracy of the results.

Few techniques have been developed to obtain high resolution results using finer grids over an area of interest in a global model instead of running a full high resolution global model. Some of the techniques are grid stretching and localized grid nesting. Grid stretching consists of refining the global resolution over an area of interest one one side and coarsen the grids on the other side by means of applying geometric function to the global cubed sphere grid. Grid nesting allows adding an additional finer grid spanning over the area of interest. The nested grid BCs are frequently updated from the coarse solution,





thus, minimizing error propagation and solution contamination compared to regional models. In addition, the two way fine-to-coarse feedback averages the fine grid solution on the nest which replaces the coarse grid solution in the region where the two grids overlap, thus, improving the solution on the coarse grid as well. For a summary of the advantages and drawbacks of grid nesting compared to grid stretching, refer to the discussion in Harris and Lin (2013).

Single grid nesting in Geophysical Fluid Dynamics Laboratory (GFDL) Finite-Volume Cubed Sphere dynamical core (FV3)
was first developed by Harris and Lin (2013). The authors ran a series of idealized tests and showed that numerical artifacts generated by the nested grid BC are comparable to those at the edges of the cubed sphere grid and decrease with increasing resolution. They also showed that the distortion of large-scale balanced flows is limited to a factor of 2 at most in global error norms when a nested grid is introduced into a global run.

Several studies were performed thereafter using the nesting capability in FV3. Harris and Lin (2014) investigated the FV3
grid nesting algorithm in GFDL's High Resolution Atmospheric Model (HiRAM). They performed simulations over the maritime continent and North America at different resolutions for both the global and nested grids. They found that two-way nesting produced less numerical errors at the grid boundaries compared to one-way nesting. They also found that orographically forced precipitation was captured in greater details and less biases were found for tropical precipitation when nesting was used. In addition, they came to the conclusion that the increase in resolution from grid nesting can by itself improve aspects
of the simulation and not because the nested grid physical parameters could be tuned independently from those of the coarser grid. Hazelton et al. (2018a) used a FV3 powered model with GFS physics, with stretched grid and a 2-km nest covering the western North Atlantic to analyze Tropical cyclones tracks, intensity and finescale structure. They compared their numerical results to observational airborne Doppler radar dataset. They found that the nested model successfully captured some structural metrics; however, some biases were found in some cases. Hazelton et al. (2018b) found that a high resolution nested model,
with a global uniform grid and a 3-km nest spanning from Africa to the western Gulf of Mexico was able to yield a performance similar to the operational GFS and other HWRF models in forecasting TC track, structure and intensity. Gao et al. (2019a) used GFDL's High Resolution Atmospheric Model (HiRAM) and showed that two-way nested model present higher forecast skills in predicting major hurricane frequency and accumulated cyclone energy compared to a nest free model. Gao et al. (2019b) investigated the two-way nesting capability of HiRAM with a 25km global resolution and a 8km nest over the tropical North
Atlantic for a set of hurricanes simulations. They compared their results to a global nest free run and observational results and found that two-way nesting yielded a better representation of several hurricane properties such as intensity and intensification rate.

Building on the single nest algorithm of Harris and Lin (2013), we present, in this paper, the multiple same level and telescoping two way nesting capability implemented in the GFDL FV3 dynamical core. This capability is supported by the
Flexible Modeling System (FMS) of GFDL and could be used within global and regional frameworks. In the following sections, we present an overview of the nesting methodology in the dynamical core, then its usage in the atmosphere model, System for High-resolution modeling for Earth-to-Local Domains (SHiELD), first in a global setup mainly focusing on the landfall of Hurricane Laura, then, in a regional setup focusing on an atmospheric river hitting California. Later, we discuss the timing and code performance.





## 2   Model description

### 2.1   Dynamical core FV3

The nonhydrostatic Finite-Volume Cubed-Sphere Dynamical Core (FV3) developed at the GFDL is used as a base of many atmospheric models for a wide range of applications from short-term weather forecasts to century long climate simulations, targeting on hurricane forecasts, chemical and aerosol transport modeling, cloud-resolving modeling, and so on. FV3 solves the non-hydrostatic compressible Euler equations on equiangular gnomonic cubed-sphere grid with a Lagrangian vertical coordinate. The horizontal grid cubed sphere geometry follows. The algorithm is fully explicit except for fast vertically propagating sound and gravity waves which are solved by the semi-implicit method. The long timestep of the entire solver is called $dt\_atmos$ which also corresponds to the physics timestep. The number of vertical remapping loops for each $dt\_atmos$ is defined by $k\_split$ where subcycled tracer advection is also performed. The acoustic timestep is defined by $n\_split$ per remapping loop yielding an acoustic timestep of $dt\_atmos/(k\_split \text{ x } n\_split)$ where sound and gravity wave processes are advanced and thermodynamics variables are advected. The detailed description of the solver horizontal and vertically Lagrangian discretizations can be found in Lin and Rood (1996, 1997) and Lin (2004).

Two-way concurrent single grid nesting was developed by Harris and Lin (2013). In this paper, we extend the single nest capability to support multiple same level and telescoping nest in global and regional domains. A nest is defined as a regional or finer grid embedded within a parent or a coarser grid. A telescoping nest is defined as a nest within a nest. The new multiple same level and telescoping nesting will be discussed in the following sections.

### 2.2   SHiELD

The System for High-resolution modeling for Earth-to-Local Domain (SHiELD) described in Harris et al. (2020) is a atmospheric research model developed at GFDL. SHiELD uses FV3 as its dynamical core, the Flexible Modeling Suite (FMS) as its infrastructure and computational framework. The physics parameterizations were originally adopted from the Global Forecast System (GFS) physics package but have been heavily updated. Currently, we use the GFDL microphysics scheme (Zhou et al. (2019)), the Eddy-Diffusivity Mass-Flux (EDMF) boundary layer scheme Zhang et al. (2015), the scale aware SAS of Han et al. (2017), the Noah Land Surface Model of Ek et al. (2003) and a modified version of the Mixed Layer Ocean of Pollard et al. (1973). Three major SHiELD configurations are being heavily tested and updated continuously: (a) Global 13-km SHiELD; (b) T-SHiELD with a static, 3-km nest spanning the tropical North Atlantic for tropical cyclones forecasts; (c) C-SHiELD with a 2.5-km nest over the contiguous United States (CONUS) for severe weather storms. SHiELD could also be configured differently depending on the application of interest, e.g. S-SHiELD for seasonal to sub-seasonal prediction is being developed. It is worth noting that all these configurations use the same codebase, pre/post-processing tools, executable following the philosophy of unified modeling "one code, one executable, one workflow". In what follows, we set up several new configurations inspired by the SHiELD family to present the new multiple nesting capability in the dynamical core FV3.



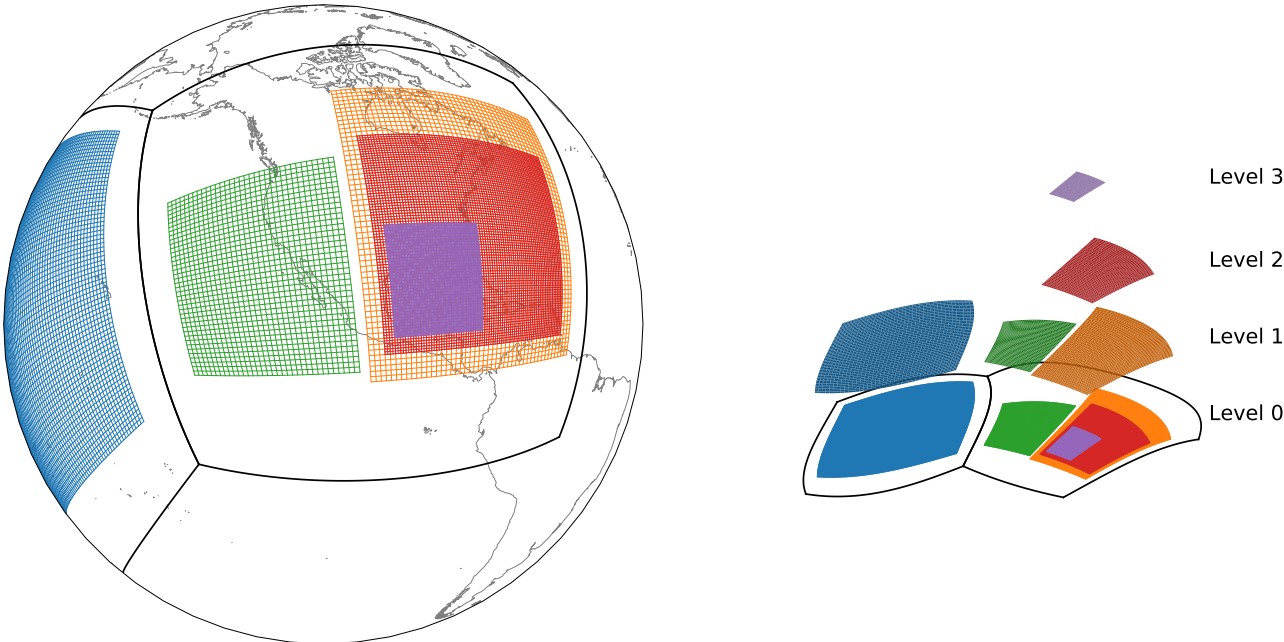

**Figure 1.** Left: Schematic showing five nests in two tiles of the top level parent grid in a global cubed sphere configuration C48_5n2. Right: Nest distribution across the levels: three nests on level 1, one nest on the second level and on nest on the third level. Nests are then projected on level 0 showing their respective locations on two parent tile of the cubed sphere.

## 3 Nesting Methodology in FV3

### 3.1 Multiple nests

Building on the single nest implementation developed by Harris and Lin (2013), the nesting capability in FV3 is extended to support multiple same level and telescoping nested grids. These capabilities are fully functional within GFDL's Flexible Modeling System (FMS). A telescoping nest is defined as a nest within a nest. A global or regional grid is considered to be at level zero and is called a top grid. A nest in one of the tiles (or tile) of the top grid (the grid at level zero) is considered to be a level one nest. A nest within the level one nest is considered as a level two nest (Telescoping nest). There is no limit on the number of nests at a particular level (for instance, we can have multiple nests at level one) and no limit on the number of levels as well. The nested grids are independent at the moment, meaning that there is no communication between nests at the same level. The communication occurs only between a child grid (nested grid) and its parent. A grid is considered as a parent grid if it holds a nest which is considered as a child grid. For a telescoping case, in the example mentioned above, the nest at level one is a parent grid relative to the nest at level two, but a child grid relative to the grid at level zero. So, a nest could be both a parent and a child grid at the same time. The nests at the same level can overlap (with no direct communication whatsoever) but should stay within their parent tile. The communication between the nests and their parents is done per level. For instance, all





nests present at a certain level (ex: level one) get their boundary condition data collectively from their parent level (level zero).
For one-way updates, the updates occur sequentially by the level number, from top to bottom (level 0 to level 1, then level 1 to
level 2, etc.) For two-way coupling, the updates occur in the opposite direction from last to second grid from last to top level.
The time-stepping on the nested grids is executed concurrently on different sets of processors and the numerical parameters on
each grid could be set differently and independently. Consequently, all these features make nesting a powerful and flexible tool

tailored to be efficiently used on parallel supercomputers.

### 3.2    Child to parent grid communication

The nesting methodology is described in details in Harris and Lin (2013) and will be summarized here. For each coarse cell
correspond $rf^2$ ($rf$ stands for refinement ratio) fine cells divided evenly in both horizontal directions. In the vertical direction,
the parent and child could have different number of vertical level[1]. As discussed in the previous section, grid communication

is only performed between a child grid and its parent. This could be roughly summarized in two steps:

1. All variables are spatially linearly interpolated from the parent grid to the nest ghost cells forming the nest boundary
conditions. These boundary conditions are updated from the parent grid each remapping time step ($dt\_atmos/k\_split$).
The nest boundary conditions are also linearly extrapolated in time from two previous coarse grid solutions for every
acoustic time step to allow the boundary conditions to evolve in time during the concurrent time stepping while waiting

for the next boundary conditions from from the parent grid at the next remapping time step. Linear interpolation processes
in not conservative by nature which may introduce some artifacts in the boundary conditions. This is a matter of future
investigation.

2. The two-way updates consists of averaging the fine grid data on the nest which will then replace the coarse grid data in
the region where the child and parent grid overlap every large time step $dt\_atmos$ before updating the physics (this was

found to maintain numerical stability). The fine scalar variables are area-weight averaged over the overlap area and then
replaces the scalar in their parent coarse grid cell. The fine staggered variables are length-weight averaged only for the
fine cells whose boundaries coincide with their parent coarse grid allowing vorticity conservation. At the moment, only
the temperature, surface pressure and the three wind components are used for the two-way updates. Therefore, there is no
violation of mass conservation during this process on the coarse grid. In addition, since the air mass is different on both

grids and since it determines the vertical coordinates, the nested averaged data is remapped to the coarse grid vertical
coordinates, meaning that it is interpolated from its fine vertical coordinate to the parent coarse vertical coordinate.

### 4    Global nesting - Hurricane Laura

We consider simulations of Hurricane Laura landfall in Louisiana (East cost) of August 2020 using GFDL's model SHiELD
(Harris et al. (2020)) for both, low and high, global uniform resolutions with different nest layouts and refinement ratios as

---

[1]Refer to the FV3 documentation for more information





shown in table 1. The initial conditions of all cases are considered at 12Z of August 26, 2020 and all times shown thereafter are

considered from this starting date. In the naming convention CA_BnC used thereafter, A refers to the number of grids per one

tile per horizontal direction of the cubed sphere (there are AxAx6 total cells on the cubed sphere grid), B the number of nests

and C the nests corresponding refinement ratios. All nests of a same case have the same refinement ratio but the refinement

ratio could be set differently for different cases. The size of the nest domains and the simulation computational cost are shown

in table A1.

| Case | Global Resolution (km) | Nests Number per Level | Nest Resolution (km) |
|---|---|---|---|
| C48 | 200 | - | - |
| C48_3n2 | | 3 Level1 | 100 |
| C48_4n2 | | 3 Level1 | 100 |
| | | 1 Level2 | 50 |
| C48_5n2 | | 3 Level1 | 100 |
| | | 1 Level2 | 50 |
| | | 1 Level3 | 25 |
| C48_4n4 | | 3 Level1 | 50 |
| | | 1 Level2 | 12.5 |
| C768 | 13 | - | - |
| C768_1n3 | | 1 Level1 | 4.3 |
| C768_2n3 | | 1 Level1 | 4.3 |
| | | 1 Level2 | 1.4 |

**Table 1.** Simulation names and details. Global resolution corresponds to the resolution of the global grid or the top level six tiles. The number of nests and their corresponding resolution is shown per level for each of the cases. Multiple nests per level have the same resolution.

### 4.1 Low Resolution global cases C48

Simulations were performed using a global gnomonic grid C48 yielding an average global resolution of 2° with an average

grid cell of 200km for the top grid. Nests of a constant refinement ratio were introduced in the following configuration: Two

nests located in tile 2 which was shifted and rotated to cover north America and one nest in tile 6 to cover part of the Pacific

Ocean as can be seen in the figure 2a. The nests in tile 2 are laid out to respectively cover the west and east coasts. All of these

nests are located at the first level, meaning that the communication occurs directly with their corresponding parent tiles. We

will refer to this case as C48_3n2 where the corresponding nest resolution is approximately ∼100km as shown in table 1 and

figure 2(a).

For case C48_4n2, an additional nest is embedded in the nest of the east coast. The additional nest is, thereby, a level two

nest or a telescoping nest whom the parent grid is the aforementioned east coast nest. The resulting telescoped nest resolution




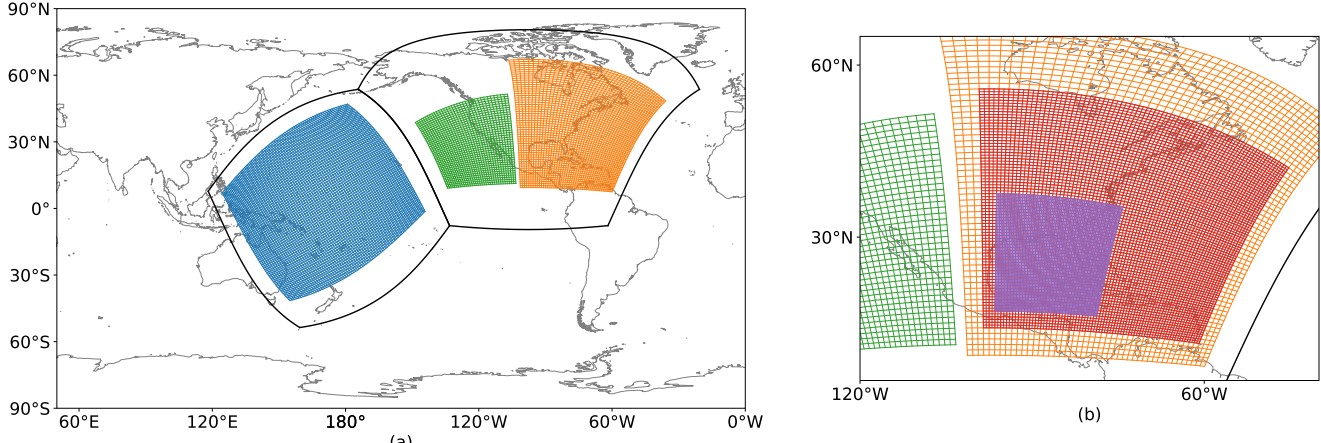

**Figure 2.** (a) C48_3n2 grid layout showing the two nests over the east and west coast in tile6 and a third nest in the pacific in tile2 (b) C48_5n2 grid layout showing the two additional (to case C48_3n2) telescoping nests nested in the east coast nest. Grid size shown in both case represents the actual real size of the grid

is ~50km. It is important to emphasize that the communication of the level two nest occurs with its parent nest of level one and not with the top grid at level0 which is the six tile cubed sphere grid. Therefore, the one-way and two-way updates of the level2 nest as described in the previous section occurs with the level one nest and at the same time the corresponding updates of the level1 nest occur with the top level0 grid.

For case C48_5n2, one more nest (a fifth nest) is added to the fourth nest of C48_4n2 as shown in figure 2b. The fifth nest yields an approximate resolution of ~25km as shown in table 1. The corresponding three telescoping layout is shown in figure 2(b).

Figure 3 shows the global 850mb vorticity field on the top level grid or level0 for cases C48, C48_3n2, C48_4n2 and C48_5n2. Left and right columns correspond to 13h and 37h after the initial date 12Z of August 26. As can be seen, Hurricane Laura is completely dissipated for case C48 which is expected since the resulting grid is too coarse (200km) to capture any details of the event.

Considering the nested cases C48_3n2, C48_4n2 and C48_5n2, the feedback from the nest in the pacific and the nest on the west coast to the top grid lead to finer structures present at those locations compared to the raw case of C48. Coming back to Hurricane Laura, C48_3n2 is able to capture its landfall with a nest resolution of 100km; however, this resolution is still too coarse as we can see from the diffusive structure of the hurricane which starts to spread on a wider area and does not present a compact structure like the one seen in cases C48_4n2 and C48_5n2.

Figure 4 shows the global 850mb vorticity field on the top level grid for cases C48 and C48_4n4. Rows from top to bottom correspond to 1h, 6h, 12h and 36h after the initial date 12Z of August 26. C48_4n4 is similar to the previous nested case C48_4n2 but with finer nests: the refinement ratio is equal to 4 for all four nests which yields a resolution of 50km on for the three level one nests and a resolution of 12.5km for the telescoping nest level2 on Laura's landfall. From the 850mb





vorticity field, we can clearly see finer structures in the region where the nest and parent grid overlap due to the two way coupling. C48_4n4 is able to capture the landfall of Hurricane Laura, the evolution of the intense weather activity in the gulf of Tehuantepec and the initial stages of formation of Typhoon Maysak in the pacific. The circles point to the weather activity in Mexico and Typhoon Maysak. It is worthy to note that C48 was able to capture typhoon Maysak but the initial stages of its

formation are not captured in details as in the case of the higher resolution nest of C48_4n4 over that region. The feedback from nests of a high refinement ratios is more pronounced compared to lower refinement ratio nests as can be seen on the west cost and the pacific regions of case C48_4n4 compared to cases C48_3n2, C48_4n2 and C48_5n2.



**Figure 3.** Global domain 850mb vorticity time evolution showing Hurricane Laura's landfall. Left column corresponds to 13h, right column corresponds to 37h for cases C48, C48_3n2, C48_4n2, C48_5n2 from top to bottom. Black lines represent nest boundaries.



**Figure 4.** Global domain 850mb vorticity time evolution showing Hurricane Laura's landfall. Left column corresponds to case C48, right column corresponds to case C48_4n4. Top to bottom rows corresponds to times 1h, 6h, 12h and 36h. Darker colors represent higher vorticity. The circles point to the weather activity in west Mexico and Typhoon Maysak. Black lines represent nest boundaries.

Hurricane Laura landfall on the first level nest (resolution 100km) is shown in figure 5 for cases C48_3n2, C48_4n2 and C48_5n2 at times 12h, 24h and 36h. As previously discussed, the hurricane structure in case C48_3n2 is more diffusive than 180 the other cases as seen in the plots of the first row. On the other hand, owing to the two-way feedback of the higher resolution





nests of cases C48_4n2 and C48_5n2, *the corresponding hurricane structure presents a more compact form and finer details are seen in the regions where the nested and parent grids overlap.*

The hurricane structure on the first level nest is very comparable for cases C48_4n2 and C48_5n2 as can be seen by in the plots of the last two rows; however, some differences still exist in that area due to the additional nest of case C48_5n2. Indeed,

looking at figure 6 which shows the hurricane evolution on the second level nest of resolution 50km, the hurricane structure is very comparable in both cases with more detailed finer structures in case C48_5n2 owing to the two-way feedback coming from the third level nest of resolution 25km.



**Figure 5.** First level nest 850mb vorticity time evolution showing Hurricane Laura's landfall. Top to bottom rows correspond to cases C48_3n2, C48_4n2, C48_5n2, respectively. Left To right correspond to 12h, 24h and 36h. Black lines represent nest boundaries.





**Figure 6.** Second level nest 850mb vorticity time evolution showing Hurricane Laura's landfall. Top to bottom rows correspond to cases C48_4n2, C48_5n2, respectively. Left To right correspond to 12h, 24h and 36h. Black lines represent nest boundaries.

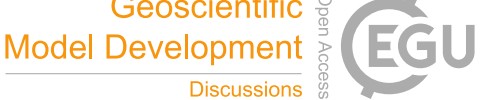

**Figure 7.** Finest grid surface wind speed time evolution showing Hurricane Laura's at time t=9, 18 and 27h of cases C48 (200km), C48_3n2 (100km), C48_4n2 (50km), C48_5n2 (25km) and C48_4n4 (12.5km). White lines correspond to velocity contours starting at 30 $m/s$ with an increment of 5

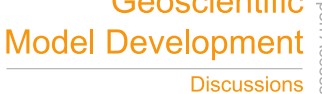

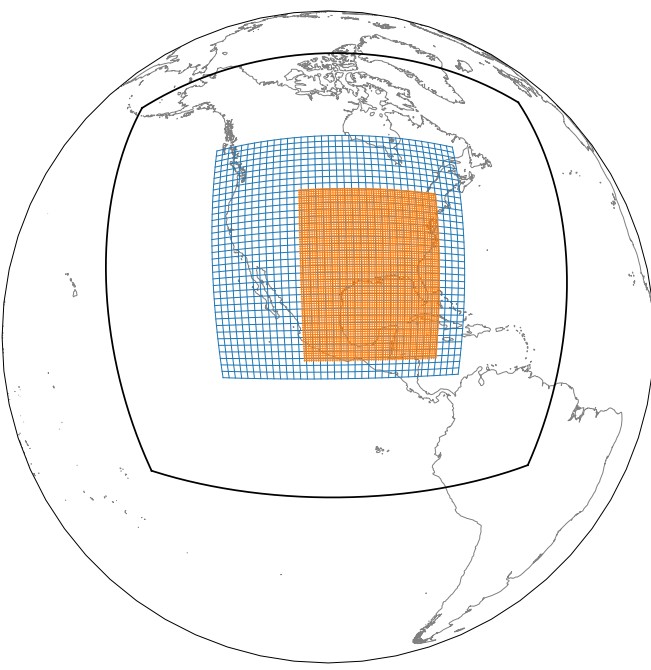

**Figure 8.** C768_2n3 grid layout showing the location of the level one and level two telescoping nests. The boundaries of the top parent tile are shown. Number of cells of the nested grids is reduced by a factor of 900.

Figure 7 shows snapshots of the 2D surface wind on the finest grid of all low resolution global cases C48 (200km), C48_3n2 (100km), C48_4n2 (50km), C48_5n2 (25km) and C48_4n4 (12.5km) at times 9h, 18h and 27h. In accordance with the vor-
ticity plots discussed earlier, decreasing the resolution results in a diffusive storm structure, while increasing the resolution by using multiple level nests and varying the refinement ratio yields a more detailed description of the storm evolution. Coarse resolutions down to 100km are incapable of capturing any of the hurricane structures while finer resolutions of 50km down to 12.5km such as in the cases of C48_4n2 (50km), C48_5n2(25km) and C48_4n4(12.5km) were capable of gradually capturing the location of the hurricane structures such as the eye and high winds region. In addition, since more details are captured
with increasing resolution, the hurricane intensity increases with nesting levels as can be seen by the contour plots (starting at 30 $m/s$) of the surface wind speed. The wind contours offsets become smaller, closer to the eye with increased resolution indicating the presence of stronger winds in that region.

## 4.2 High Resolution global cases C768

Simulations of the same event, Hurricane Laura, were performed with a higher global resolution of C768 yielding an average
grid width of 13km for the global parent grid. In addition, cases with one nest, C768_1n3, and two nests, C768_2n3, centered on the east coast over the hurricane were considered. The C768_2n3 is simply a C768_1n3 with a telescoping nest: a level two





nest is embedded in the level one nest as shown in figure 8. The resulting grid resolution of the nests is, consequently, 4.3km for the first level nest and 1.4km for the telescoping nest as shown in table 1.

Figure 9 shows the time evolution of 2D surface wind speed for on the finest grid of cases C768, C768_1n3, C768_2n3 at

9h, 18h and 27h with a corresponding approximate resolution of 13km, 4.3km and 1.4km from top to bottom. Finer details are captured when going to higher resolution from top to bottom rows similar to the results of the previous section. This is also demonstrated by the non-smoothness of the contour plots around the storm center. It is also worth noting that the storm travels slower in the high resolution case as seen by the location of the eye: the eye reaches the cost faster in case C768_1n3 compared to C768_2n3 which is a matter of future investigation.





**Figure 9.** Finest grid surface wind speed time evolution showing hurricane Laura's at time t=9, 18 and 27h of cases C768 (13km), C768_1n3 (4.3km), C768_2n3 (1.4km), white lines correspond to velocity contours starting at $30 m/s$ with an increment of 5.





**Figure 10.** Column-integrated water vapor, PWATclm, hourly-accumulated precipitation, PRATsfc, hourly-accumulated precipitation due to convection parameterization, CPRATsfc, all shown at time 18h on the finest grid of cases C768, C768_1n3 and C768_2n3. Colorscale selected to capture light and heavy rain.





Figure 10 shows the column-integrated water vapor, the hourly-accumulated precipitation and the hourly-accumulated precipitation due to convection parameterization at t=18h on the finest grid of C768, C768_1n3 and C768_2n3. Similar to the velocity field results, finer details are captured with increasing resolutions. The intensity of rain seems to increase with increased resolution and becomes more spread out when looking at the 1km nest. The position of these small scale features of high intensity indicates the geographic location where most of the storm damage is likely to occur. On the other hand, the

precipitation due to convection parameterization decreases in intensity with increasing resolution. This is expected since higher resolution will resolve the bulk of the precipitation and thus the contribution coming from parameterization will be minimal and the contribution of the resolved precipitation will be maximal as previously mentioned. Note that the heaviest rain rates even at 13-km resolution are from the resolved-scale precipitation. The right column of figure 10 also clearly shows the scale-awareness of the SAS convection scheme Han et al. (2017) as it becomes less active at higher resolutions, an valuable asset for

our multi-scale modeling.

### 4.3   Quantitative analysis

Figure 11 shows the time evolution of (a) maximum surface wind speed, (b) the minimum sea level pressure, (c) 500mb vertical updraft of the finest nested grid of each of the low and high resolution cases shown in table 1. For instance, for case C768_2n3, we show the evolution on the level two nest of a 1.4 km resolution; for case C48_4n4, the results of the 12.5km nest are shown

and so on. The lower resolution cases C48 and C48_3n2 present constant values for the three variables as time advances. This shows that a coarse resolution down to 100km is insufficient of capturing the event. For all other cases, we notice an overall increase in storm intensity with nesting levels or with increasing resolution. The maximum surface wind speed increases in magnitude from the initial time to reach a peak of approximately 60 m/s between 10h and 20h for cases C768, C768_1n3 and C768_2n3. Then, it starts to decrease between 25h and 30h, setting the mark of the beginning of the landfall. Those three

curves almost overlap, showing that a minimum resolution of 13km might be enough to capture the maximum surface wind speed time evolution. For the intermediate resolutions, we notice an increase of surface wind speed magnitude with an increase of nesting level with a delayed peak to around 25h; however, this starts to decrease shortly after this time for all cases.

     A similar behavior is seen for the minimum sea level pressure. A minimal coarse resolution of 100km is unable to capture the event. All cases reach a minimum of 950mbar some time between 10h and 30h then decrease at the 30h mark. The sea

level pressure decreases with increasing resolution; however, we notice that the high resolution cases C768_1n3 and C768_2n3 present a slightly higher minimum sea pressure compared to cases C768, C48_5n2 and C48_4n2 which is a matter of future investigation.

     The 500mb vertical velocity increases systematically with increasing resolution to reach a value 20m/s for the highest resolution nest. As seen in the plot, the curves corresponding to resolutions coarser than 13km almost overlap, and the curves

offset starts to increase from a 13km resolution to the 1km nest. This shows that while a 13km resolution looks enough to capture an approximate trend of the time evolution of the velocity and pressure fields, a higher resolution is still required to capture to the 500mb vertical updraft.



Finally, the results of cases C48_4n4 (12.5km) and C768 (13km) are pretty similar which show that telescoping nesting in a low resolution global setup is able to capture the evolution of the event similar to a high resolution global case. In addition, the

telescoping setup is still computationally cheaper as discussed in section 6



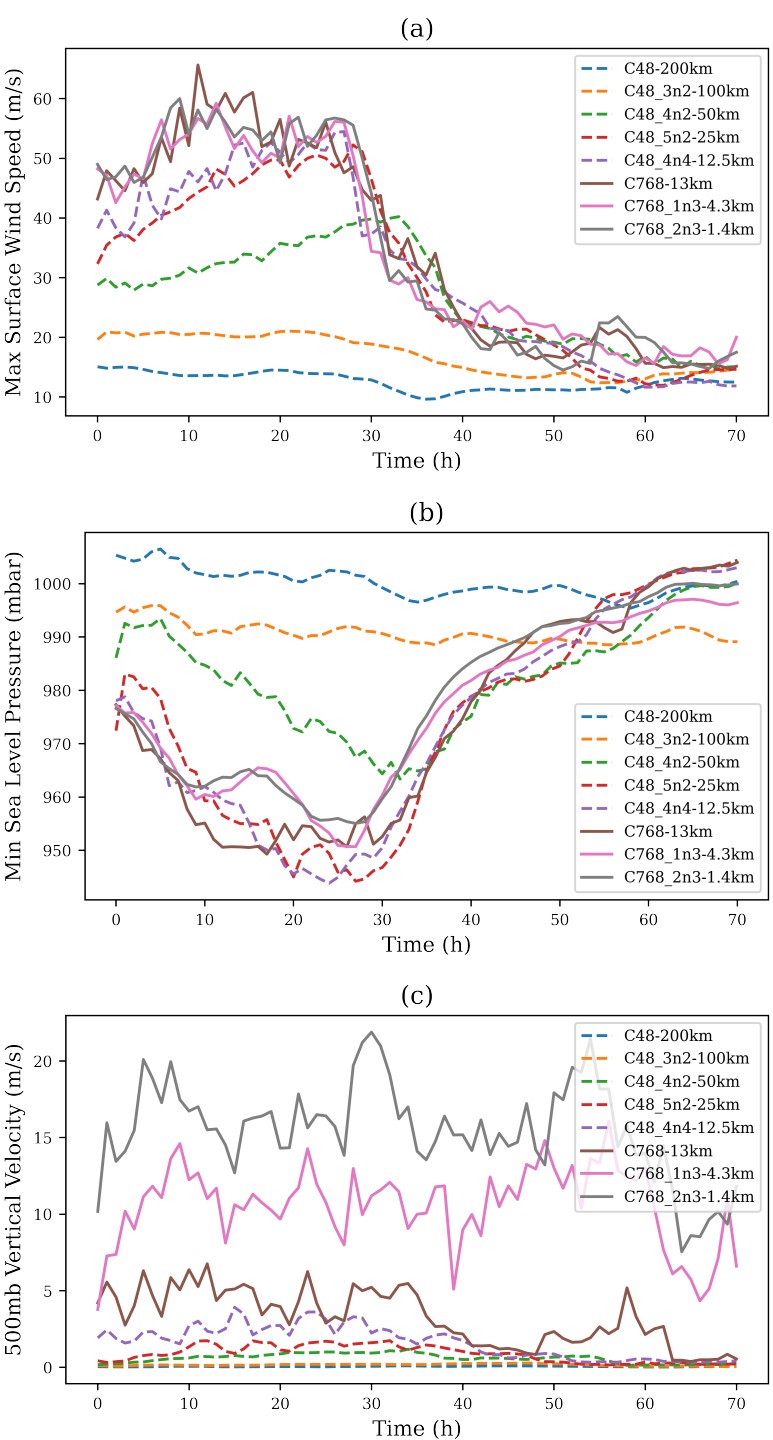

**Figure 11.** Time evolution of (a) maximum surface wind speed, (b) the minimum sea level pressure, (c) 500mb vertical updraft of the finest nested grid in each of the cases shown in table 1





## 5 Regional nesting - Atmospheric River

We now show nested-grid simulations of an atmospheric river striking the US West Coast in late January 2021 (https://cw3e.ucsd.edu/cw3e-event-summary-26-29-january-2021/) in a regional telescoping nesting framework within SHiELD. The setup consists of a regional domain spanning from the eastern pacific to the west coast, embedding two subsequent level one and level two nests as

shown in figure 12. Tile one stands for the regional tile while tile two and three correspond to the level one and level two nests. The nests refinement ratios is three corresponding to a configuration of an approximate resolution of 50-17-6km as shown in table 2. The simulations initial date is 00Z January 26, 2021. he size of the nest domains and the simulation computational cost are shown in table A1.

    Figure 12 shows the precipitation rate at t=48h on (a) tile one of R192, (b) tile one of R192 and tile two of R192_1n3 (c) tile

one of R192, tile two of R192_1n3 and tile three of R192_2n3. Results show that increasing the resolution from top to bottom allows capturing finer details in the region where the nested grids are located. Moreover, the nest boundaries at either nesting level do not introduce numerical noise in the parent solution. This is specifically shown by the continuity of the resulting rain-band patterns across the nest boundaries when overlaying the fine solution on the solution of its parent grid. Figure 13 shows the accumulated frozen precipitation during the three simulated days. The frozen precipitation is defined as the depth

of liquid water equivalent to the frozen precipitation that has fallen. From top to bottom, the results are shown, respectively, for the highest resolution grid of R192, R192_1n3 and R192_2n3. Higher resolution nests give a more detailed description of geographic distribution of the orographic frozen precipitation. In addition, enhanced resolution is able to capture more intense snowfall in some areas which is in accordance with the results of the previous section.

| Case | Regional Resolution (km) | Nests Number per Level | Nest Resolution (km) |
|------|--------------------------|------------------------|----------------------|
| R192 | 50 | - | - |
| R192_1n3 | | 1Level1 | 17 |
| R192_2n3 | | 1Level2 | 6 |

**Table 2.** Simulation names and details. Regional resolution corresponds to the resolution of the regional grid or the top level one tile grid. The number of nests and their corresponding resolution is shown per level for each of the cases.

## 6 Code timing and performance

The size of all nest domains, timestep details, number of cores and simulation time for all cases are shown in table A1. $n\_split$ was chosen to be in the interval [5,12] and $k\_split$ in a way to get an appropriate Courant number for the different grids. It is clearly obvious that a low resolution global grid with high resolution multiple nests requires less computational resources than a high resolution global grid. In fact, if one is interested in one or more particular events, such as the landfall of Hurricane Laura, using a C48_4n4 setup is much cheaper than a global C768. The telescopic nest of C48_4n4 spanning over the east





**Figure 12.** Hourly-accumulated precipitation, PRATsfc, at t=48h (a) tile one of R192, (b) tile one of R192 and tile two of R192_1n3 (c) tile one of R192, tile two of R192_1n3 and tile three of R192_2n3

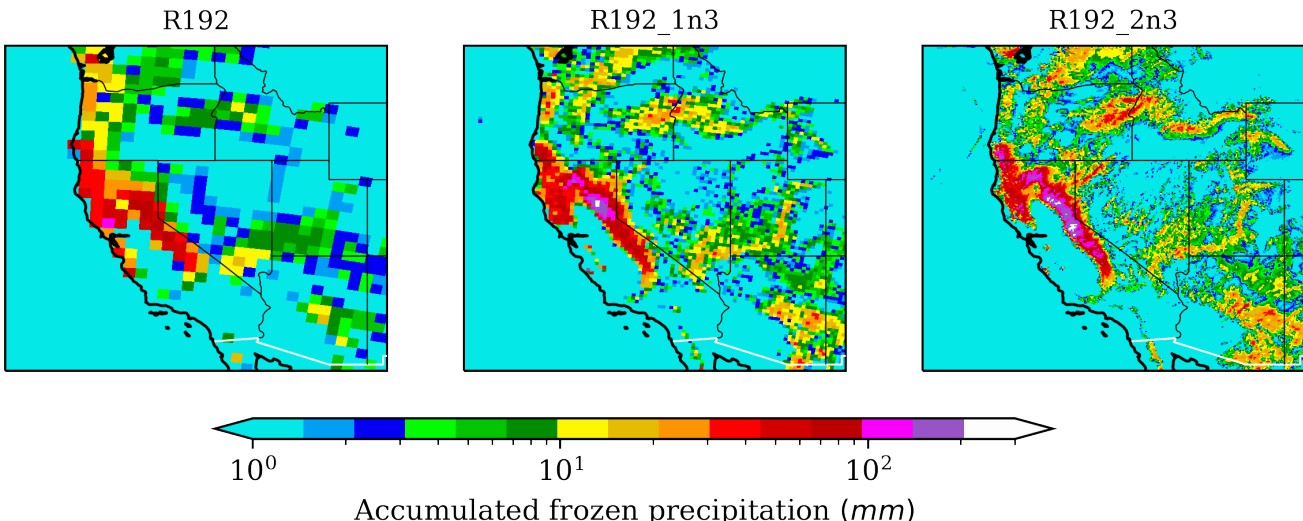

**Figure 13.** Total three days accumulated frozen precipitation on the highest resolution grid of R192, R192_1n3 and R192_2n3. Note that this is liquid water equivalent accumulation.

coast yields a resolution of 12.5km which is similar to the 13km of C768. The C48_4n4 requires 140 core-hours to simulate a three days simulation whereas a C768 case requires 1465 core-hours. This is a 10x reduction in computational resources. In addition, concurrent nesting allows the user to consider additional nests on more cores without compromising the run time of a case. This can be seen by looking at the respective timings 680s and 687s of C48_4n2 and C48_5n2 where adding a level 3 nest with 36 cores for case C48_5n2 didn't degrade the run time of the three days simulation. It is worth mentioning that not
all cases in table A1 are fully optimized to get the best computational performance.

## 7  Conclusions

We present, in this study, the multiple same level and telescoping two-way nesting capability implemented in the Geophysical Fluid Dynamics Laboratory Finite-Volume Cubed-Sphere dynamical core FV3. Simulations were performed within GFDL's weather model SHiELD with low and high resolutions multiple same level and multi-level telescoping nests under both global
and regional configurations. Hurricane Laura's landfall was simulated with both low and high global resolution setups with multiple nests with resolutions spanning from 200km down to 1km. In addition, for the low resolution global setup, nests were spread out over the globe covering different geographic locations of interest and were able to capture several independent and simultaneous weather events such as the initial stages of formation of Typhoon Maysak and intense weather activity at the Western gulf of Mexico. The multi-level telescoping nesting capability was shown to work well in capturing fine scale flow
features during the landfall of Hurricane Laura at different nest levels. The intensity of the storm increased up to a certain resolution. The velocity magnitude and sea level pressure showed a similar behavior during the landfall. Precipitation and



moisture increased with increased resolution as well. Two way nesting updates, at various telescoping levels, were shown not to introduce numerical artifacts to their corresponding parent grids at the cells where the nest boundaries overlap with their corresponding coarse cells.

Additionally, an atmospheric river event that hit California in January 2021 was simulated in SHiELD in a regional setup. Two nests forming a telescoping setup were considered. The rainfall and total three days accumulated frozen precipitation were found to increase with increased resolution. In addition, higher resolutions gave a more detailed description of the geographic distribution of these precipitations. Furthermore, the precipitation rain-bands presented a continuous pattern when crossing a nest boundary when overlaying the fine solution on the coarse one, emphasizing the lack of artifacts of the current two way

nesting algorithm.

Furthermore, concurrent multiple nesting has been shown to require less computational resources than a global setup of a comparable resolution. In addition, additional nests do not compromise the performance of a simulation setup if the number of cores is distributed among the grids in a optimal manner.

Multiple nesting is now part of the official FV3 release. Current and future efforts will focus on developing algorithms to

enable moving nests to track weather events such as tropical storms. In addition, nest spanning multiple tiles are currently supported by the infrastructure FMS and will be implemented in the dynamical core FV3.

**Appendix A**

*Code availability.* SHiELD could be built and run from the latest GitHub official releases of GFDL's Finite-Cubed Sphere Dynamical core FV3: https://github.com/NOAA-GFDL/GFDL_atmos_cubed_sphere, the Flexible Modeling Systems FMS: https://github.com/NOAA-

GFDL/FMS and Global Forecast Physics GFS: https://github.com/NOAA-GFDL/SHiELD_driver

*Competing interests.* The authors declare that they have no conflict of interest.

*Acknowledgements.* We thank Kun Gao and Jan-Huey Chen for their review and useful comments that improved the quality of the manuscript. We also thank Kai-Yuan Cheng for all the insightful discussions and valuable assistance during this project.



| Case | Grid Cells | Cores | dt_atmos | k_split | n_split | Three days simulation time (s) |
|------|------------|-------|----------|---------|---------|-------------------------------|
| C48 | 48x48x6 | 6x6x6=**216** | 200 | 1 | 6 | 83 |
| C48_3n2 | 48x48x6 | 6x6x6 | | 1 | 6 | |
| Lev1 | 71x79 47x59 35x41 | 6x6 3x3 6x6 | | 2 2 2 | 12 12 12 | 666 |
| Total | | **297** | | | | |
| C48_4n2 | 48x48x6 | 6x6x6 | | 1 | 6 | |
| Lev1 | 71x79 47x59 35x41 | 6x6 3x3 6x6 | | 2 2 2 | 12 12 12 | 680 |
| Lev2 | 81x89 | 6x6 | | 2 | 12 | |
| Total | | **333** | | | | |
| C48_5n2 | 48x48x6 | 6x6x6 | | 1 | 6 | |
| Lev1 | 71x79 47x59 35x41 | 6x6 3x3 6x6 | | 2 2 2 | 12 12 12 | |
| Lev2 | 81x89 | 6x6 | | 2 | 12 | 687 |
| Lev3 | 71x81 | 6x6 | | 2 | 12 | |
| Total | | **369** | | | | |
| C48_4n4 | 48x48x6 | 6x6x6 | | 1 | 6 | |
| Lev1 | 145x145 89x117 69x81 | 12x12 8x10 8x6 | | 2 2 2 | 12 12 12 | 570 |
| Lev2 | 317x337 | 20x20 | | 2 | 12 | |
| Total | | **888** | | | | |
| C768 | 768x768x6 | 30x30x6=**5400** | 90 | 1 | 5 | 977 |
| C768_1n3 | 768x768x6 | 14x14x6 | | 1 | 5 | |
| Lev1 | 1081x1081 | 20x20 | | 2 | 6 | 5141 |
| Total | | **1576** | | | | |
| C768_2n3 | 768x768x6 | 16x16x6 | | 1 | 5 | |
| Lev1 | 1081x1081 | 22x22 | | 2 | 6 | 15128 |
| Lev2 | 1747x2203 | 35x44 | | 5 | 10 | |
| Total | | **3560** | | | | |
| R192 | 116x93 | 6x8=**48** | 90 | 4 | 5 | 1241 |
| R192_1n3 | 116x93 | 6x8 | | 4 | 5 | |
| Lev1 | 310x241 | 12x10 | | 2 | 10 | 2206 |
| Total | | **168** | | | | |
| R192_2n3 | 116x93 | 6x8 | | 4 | 5 | |
| Lev1 | 310x241 | 12x10 | | 2 | 10 | 2650 |
| Lev2 | 376x559 | 18x18 | | 2 | 10 | |
| Total | | **492** | | | | |

**Table A1.** Number of grid cells, cores, remapping timesteps per dt_atmos, acoustic timesteps per k_split and simulation time of all runs. Number of grid cells and cores are shown on the same line for all nests at the same level. These runs could still be optimized by redistributing the number of cores among grids in a more efficient way but this is not pursued here.





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
