# Peer review of "Multiple same-level and telescoping nesting in GFDL's dynamical core"

_Geoscientific Model Development, 2021_

## Author Response (AR1)

*We would like to thank the referee for the careful consideration of our work; we also appreciate the useful comments and suggestions to help us improve the quality of our contribution. We have provided a rebuttal to the specific comments below and referred to the subsections of the corresponding changes in the new version of our manuscript. Please note that the modifications to the manuscript are written in blue; the new additions to the manuscript are in red.*

**Referee 1**

**General comments**

Multiple same-level and telescoping nesting in GFDL's dynamical core by Mouallem et al documents the implementation of the telescopic nesting technique within the FV3 dynamical core for "System for High-resolution modeling for Earth-to-Local Domains (SHiELD)" model and for potential implementation in the Unified Forecast System (UFS). The authors have systematically studied the impacts of improved resolution that could be attained via telescopic nesting on a case of hurricane Laura and nested-grid simulations of an atmospheric river striking the US West Coast. This work is well motivated and well written. I believe this is an important development for the UFS, as well. This work should be accepted for a publication. I have only some minor suggestions that the authors may wish to consider before submitting the final version.

*REBUTTAL: We would like to thank the referee for their helpful and supportive comments*

**Detailed comments**

Since this is an important document, it may be worthwhile to discuss the grid structure, grid staggering and the variables on staggered grid. A figure showing the nested grid inside the parent grid may be useful. Where and which variables are placed on A, C and D grids? How does the feedback occur? Similarly the grid structure related to boundary conditions updates may be useful.

*REBUTTAL: The averaging process of the twoway feedback of scalars and staggered variables is discussed in section 3.2/point2. Additional technical details of twoway updates as well as a figure illustrating the feedback from a fine to coarse grid can be found in the FV3 technical document section 'grid nesting'. All variables are cell-mean quantities except for the winds. For conciseness, we prefer not to add all this information to the current paper but we added text to clarify some details and refer to the technical document.*

Was the same physics used all the way from 200 km grid length (C48) to 1.4 km grid resolution (C768_2n3)? What about the horizontal diffusion and/or divergence damping coefficients for various resolutions? A table for physics along with k_split and n_split for various resolutions and perhaps other namelist changes for different grid resolutions may add more information to readers and model users.

*REBUTTAL: We agree with the referee. The timestepping parameters are shown in the appendix. We have uploaded the input namelists of the cases separately to zenodo for other readers and model users.*

The section on Atmospheric river looks little rushed. This section needs more description. Figure 12 may need improvements because it does not provide much information.

*REBUTTAL: We agree with the referee. We added more text to provide more background, motivation, and description for the reader.*

*We would like to thank the referee for the careful consideration of our work; we also appreciate the useful comments and suggestions to help us improve the quality of our contribution. We have provided a rebuttal to the specific comments below and referred to the subsections of the corresponding changes in the new version of our manuscript. Please note that the modifications to the manuscript are written in blue; the new additions to the manuscript are in red.*

**Referee 2**

**General comments**

This manuscript documented the effort and work developing and expanding the multiple same-level and telescopic nesting capabilities for the GFDL's FV3 dynamical core. Experiments were conducted for both global and regional configurations to demonstrate the effectiveness and advantages of using the multiple and telescopic nesting capabilities. Overall, the manuscript is well organized and prepared. I only have a few minor concerns (see details in the specific comments below) before the manuscript can be accepted for publication.

*REBUTTAL: We would like to thank the referee for their helpful and supportive comments about our work.*

**Detailed comments**

In terms of the multiple same level nesting, it is mentioned in Section 3.1 that, there is no limit on the number of nests at a particular level. However, it is not clear whether or not the same level nests can overlap with each other. If so, how the over-lapped areas are treated, especially when the two-way nesting feedback is turned on?

*REBUTTAL: It is mentioned in section 3.1-L103: "The nests at the same level can overlap (with no direct communication whatsoever) but are required to stay within their parent tile", and in section 3.1-L107: "For two-way coupling, the updates occur in the opposite direction from last to second grid from last to top level". No attempt is made to blend data from the overlapping nests when performing the two-way update; the two-way updates are done independently and in succession.*

Also, related to the two-way nesting feedback, it is stated (lines 127-128) that "At this moment, only temperature, surface pressure and the three wind components are used for the two-way updates." Could the author comments, why only these variables are currently considered/implemented for two-way nesting feedback? How about other prognostic variables (3-d pressure or geopotential height, 3-d microphysics tracer variables, surface variables, etc.)? In the meantime, have the authors considered/compared among different nesting settings, for example, full two-way nesting feedback vs partial (say 50%) two-way nesting feedback vs one-way nesting without feedback?

*REBUTTAL: Currently, only temperature and the winds (not surface pressure) are updated from the nested grid and the parent. This trivially ensures mass conservation of the dry air and of all tracers on the global domain, a crucial need for longer-term simulations. Results from Harris and Lin (2013, 2014) show that this does not degrade scientific performance. The smaller number of updated variables also greatly reduces the data that needs to be passed between the grids, improving model efficiency especially for simulations with complex microphysical, aerosol, or chemical schemes.*
*We have not performed a comparison of different nesting settings in this study; however, previous studies have analysed one-way vs two-way nesting (Harris and Durran 2010; Harris and Lin 2013, 2014). FV3 does allow partial two-way feedback, which is performed in the Rayleigh damping layer to reduce the effect of the*

*update in the upper levels of the parent domain. This capability may be expanded in a future release of FV3.*

**In Table A1, It looks to me that, the parent and multiple and telescopic nested domains all use the same dt_atmos (physic time step), though they may have different dynamics and acoustic time steps (when using different k_split/n_split settings). Is this (using the same dt_atmos for parent and nests) a requirement/limitation for the current nesting implementation in the FV3 dynamical core, or one could choose to use different dt_atmos values for different parent/nested domains?**

*REBUTTAL: That is correct, at the moment, all nested grids follow the dt_atmos of the top parent grid and it is indeed a requirement for the current nesting implementation. We added text to clarify this information on page5.*

**Technical comments/corrections**

**Page 1, abstract, line 4: Change "... were able capture ..." into "... were able to capture ...".**
*REBUTTAL: Done.*

**Page 4, Figure 1 caption: Correct "one nest on the second level and on nest on the third level".**
*REBUTTAL: Done.*

**Page 3, line 83: Please provide the full term for SAS.**
*REBUTTAL: Done.*

**Page 5, line 120: Fix "... the next boundary conditions from from the parent grid at the next remapping time step. Linear interpolation processes in not conservative by nature ...".**
*REBUTTAL: Done.*

**Page 5, line 133: Fix "(East cost)".**
*REBUTTAL: Done.*

**Page 22, line 252: Fix "he size ...".**
*REBUTTAL: Done.*